# Mind the (Social and Emotional Competence) Gap to Support Higher Education Students’ Well-Being: Psychometric Properties of the SECAB-A(S)

**DOI:** 10.3390/ejihpe15080162

**Published:** 2025-08-16

**Authors:** Sofia Oliveira, Tiago Maçarico, Ricardo Pacheco, Isabel Janeiro, Alexandra Marques-Pinto

**Affiliations:** 1Business Research Unit (BRU), ISCTE—Instituto Universitário de Lisboa, 1649-026 Lisbon, Portugal; 2CICPSI, Faculdade de Psicologia, Universidade de Lisboa, 1649-013 Lisbon, Portugal; tiagomacarico@edu.ulisboa.pt (T.M.); injaneiro@psicologia.ulisboa.pt (I.J.); ampinto@psicologia.ulisboa.pt (A.M.-P.); 3Faculdade de Psicologia, Universidade de Lisboa, 1649-013 Lisbon, Portugal

**Keywords:** assessment, confirmatory factor analysis, higher education students, instrument, invariance, psychometric study, reliability, social and emotional competence, validity

## Abstract

Today’s increasingly brittle, anxious, nonlinear, incomprehensible world of work calls for a socially and emotionally competent workforce. However, there is a clear gap in higher education settings regarding the assessment and promotion of students’ social and emotional competence (SEC). Our study aims to address the pressing need to evaluate and develop higher education students’ SEC by providing a tool to assess these skills, enabling researchers and practitioners to intervene and actively promote them. A sample of 767 higher education students (62.8% female, *M* = 22.88 years, *SD* = 7.30) enrolled in the study. Structural, discriminant and concurrent criterion validity, and reliability of the measure were assessed. A multiple hierarchical regression analysis tested the relation of SEC and well-being. Confirmatory Factor Analysis supported the hypothesized factorial structures. Coefficient omegas indicated adequate internal consistency. The results also supported the measure’s discriminant and criterion validities in relation to external measures. Multi-group invariance across gender and academic fields was attained. We found evidence of the predictive role of intrapersonal skills on students’ personal and academic well-being. This study bridges a gap in research and practice by introducing a psychometrically sound yet parsimonious instrument for assessing higher education students’ SEC. It also highlights the supportive role of SEC in promoting students’ well-being.

## 1. Introduction

The rapid evolution of the job market, driven by forces such as globalization, the emergence of novel professions, and transformative technological advancements like Artificial Intelligence, leads to a future characterized by significant uncertainty ([60]). These rapid transformations in the world of work also lead to elevated levels of workplace stress. This intensification of pressure stems from several key factors, including the blurring of work–life boundaries inherent in the “always-on” culture fueled by 24/7 digital connectivity, heightened demands for productivity driven by global competition, and the increasing cognitive load associated with greater job complexity ([38]; [60]; [106]). Consequently, navigating this unpredictable landscape makes it essential to develop core human skills such as adaptability, complex problem-solving, self-awareness, behavioral and emotional regulation, effective teamwork, and robust communication ([23]; [30]; [37]; [60]; [107]).

Although crucial, these skills are not always explicitly taught in academic contexts. Instead, from the classroom to professional training, their cultivation often occurs through more indirect, relational means within educational settings ([14]; [98]). According to the [77] ([77], [78]), students who develop positive relationships with both their teachers and peers tend to show stronger social and emotional skills. Supportive peer-to-peer interactions are associated with higher levels of trust, optimism, and sociability. Likewise, positive student–teacher relationships contribute to increased motivation, persistence, and curiosity, all of which play a crucial role in students’ overall development and academic success. These findings have led to a consensus among researchers, practitioners, and policymakers regarding the importance of assessing and developing social and emotional skills across the lifespan ([30]). However, existing efforts have predominantly concentrated on children and adolescents ([30]), leaving a significant gap in both research and practice concerning adult populations ([81]). This gap is even more pronounced within the context of higher education, as students in this transitional stage are often overlooked in both youth- and adult-focused research ([22]). Our study aims to address this need by providing a tool to assess higher education students’ Social and Emotional Competence (SEC), enabling us to intervene and actively promote it.

### 1.1. Social and Emotional Learning’s Impact on Students’ Mental Health and Well-Being

Mental health and well-being concerns among young people have long been recognized ([111]); however, data collected since the COVID-19 pandemic indicate that children and adolescents under 18 years of age (who are currently most of our higher education students) have experienced a disproportionately severe impact compared to other age groups ([76]). Moreover, subsequent global developments following the pandemic, such as armed conflicts, economic downturns, and sociopolitical instability, potentially exacerbate risks to mental health and well-being, particularly among younger populations ([23]; [76]). These circumstances underscore the critical need for intensified efforts in knowledge generation, research, and intervention strategies within this domain.

Studies in school contexts have consistently shown that students with stronger emotional regulation and interpersonal skills are more likely to adopt healthy behaviors, have increased school achievement ([20]; [31]), and report higher levels of well-being and mental health ([103]). Social and Emotional Learning (SEL) has therefore become a central focus of numerous interventions, especially within preschool, primary, and secondary education ([103]). Informed by the Emotional Intelligence Theory ([92]), which defined emotional intelligence as the “ability to monitor one’s own and others’ feelings and emotions, to discriminate among them and to use this information to guide one’s thinking and actions” ([92]), SEL refers to the process through which both children and adolescents develop and effectively use the knowledge, attitudes, and skills necessary to regulate their emotions, set and accomplish positive goals, show empathy toward others, build and maintain healthy relationships, and make responsible choices ([30]). More specifically, [110] ([110]) have defined SEC as a core set of cognitive, emotional, and behavioral skills individuals develop through SEL processes ([35]). These competencies include the ability to understand and manage one’s own emotions (self-awareness and self-regulation), demonstrate empathy and perspective-taking (social awareness), build and sustain positive interpersonal relationships (relationship skills), and make responsible, ethical decisions (responsible decision-making) ([22]; [31]; [33]; [35]).

Together, these skills support adaptive functioning across academic, social, and personal domains, and they have been promoted in school settings worldwide for children and adolescents over the past three decades. Research has shown that evidence-based SEL programs yield long-lasting positive outcomes in behavioral, attitudinal, emotional, and academic areas (e.g., [32], [31]; [63]; [75]; [105]). More broadly, SEL fosters harmonious relationships, social cohesion and inclusion, positive attitudes toward diversity, equity, and social justice, as well as improved mental health and well-being among children and adolescents ([19]). However, while a strong foundation is laid during these earlier development stages, SEL is equally relevant in higher education ([31]), where further research and intervention efforts are needed ([88]).

### 1.2. SEL in Higher Education

Higher education students are widely considered a vulnerable group regarding mental health and well-being, as emerging adulthood ([6]) is a developmental stage marked by the dual tasks of consolidating identity and forming close, meaningful interpersonal relationships ([36]; [64]). This developmental trajectory intersects with the transition to higher education, a period marked by distinct academic, emotional, and social demands—including increased workload, greater personal responsibility, time management difficulties, and experiences of social isolation ([16]; [22]; [41]). Successfully navigating these transitions often requires students to re-establish their social networks and engage in the exploration of romantic relationships ([28]; [104]), while also adopting self-directed, deep learning strategies ([11]). These challenges are compounded by a shifting career landscape characterized by instability, multidirectional mobility, and the intersection of multiple life roles ([62]).

Faced with increased demands, higher education students often experience stress, anxiety, and difficulties adjusting to academic life, which can negatively affect their academic performance, mental health, and overall well-being ([17]; [23]), thereby highlighting the critical need for the continued development and application of SEC during this critical period ([22]; [33]). Intrapersonal skills, such as self-awareness, help students understand and regulate their emotional responses to stressors like academic pressure. Self-regulation is equally vital, enabling students to juggle competing demands, stay organized, and manage academic deadlines, social life, and extracurricular commitments. Notably, self-control, as a specific skill within the self-regulation domain, has been shown to be a significant predictor in preventing college dropout ([29]). At the same time, interpersonal competencies play a pivotal role, with higher education students experiencing a heightened need for social awareness and relationship skills ([22]). The transition to higher education often involves forming entirely new social connections, making relationship skills essential for building supportive networks with peers, professors, and colleagues—networks that contribute not only to academic adjustment but also to mental health and well-being ([17]; [22]). In addition, social awareness becomes increasingly relevant in the diverse university context, where respectful engagement with different cultures, perspectives, and identities is key to fostering inclusion and positive interaction ([22]). Finally, responsible decision-making supports ethical judgment and supports students’ capacity to engage in open-minded, thoughtful, informed, and socially responsible choices across academic and extracurricular domains ([23]). In sum, a growing body of empirical research has provided evidence supporting the relationship between higher education students’ SEC and their mental health and well-being, academic engagement, academic performance, and college completion (e.g., [1]; [23]; [94]; [112]).

Although traditionally overlooked in higher education ([88]), a growing body of research highlights the association between higher education students’ social and emotional adjustment and both academic achievement and persistence ([22]; [33]). These findings underscore the importance of extending SEL initiatives to higher education settings, where SEC plays a vital role in supporting students’ successful adaptation and enhancing their ability to navigate complex challenges ([22]; [88]). Importantly, acquiring social and emotional skills does more than support immediate academic success; it lays the foundation for lifelong learning. These competencies contribute to the development of higher-order thinking skills, employability skills, and civic, consumer, and life skills ([33]). Fostering social and emotional development in higher education is essential—not only for helping students thrive academically and personally during their studies and for preparing them to navigate future personal and professional challenges, but also for promoting their mental health and overall well-being. Evidence indicates that proficiency in SEC increases the likelihood of higher education students adopting healthier lifestyle habits, achieving academic success, and performing effectively in professional contexts ([88]). Thus, higher education students who are better prepared to adapt to the academic, social, and emotional demands of higher education report more positive mental health outcomes ([17]; [88]). In particular, social connections and a sense of belonging, both in peer relationships and in the context of developing romantic relationships ([28]), play a crucial role in this adjustment process, helping students avoid social isolation and loneliness, which are known predictors of stress, anxiety, and depression ([17]). This need for connection and belonging reflects the basic psychological need for relatedness, a core component of the Basic Psychological Needs Theory (BPNT, which integrates the broader Self-Determination Theory; [90]). The need for relatedness reflects the desire to feel connected, supported, and significant to others ([26]; [90]). By promoting empathy, collaboration, and effective communication skills ([30]), interpersonal SEC directly contributes to meeting this psychological need, helping individuals experience authentic relationships and social support ([57]). Research has also shown that satisfying higher education students’ need for relatedness to peers and faculty predicts academic outcomes relevant to job preparation ([8]). Although interpersonal skills play a pivotal role for higher education students ([22]), being more directly associated with the need for relatedness, BPNT posits that optimal functioning and psychological well-being also require the satisfaction of two additional needs: autonomy and competence ([26]; [90]). These three needs are closely aligned with the five key domains of SEC ([57]). SEL can be seen as a practical educational method that helps meet these needs: it promotes students’ autonomy by fostering self-awareness and responsible decision-making, enhances competence through goal-setting and self-regulation, and fosters relatedness by cultivating social awareness and relationship skills ([57]). In sum, BPNT offers a coherent theoretical lens to understand how SEL can foster higher education students’ personal growth and emotional development by ensuring a learning environment that supports their basic psychological needs ([57]). When these needs are satisfied, they not only protect against higher education students’ demotivation but also foster greater academic engagement and promote students’ life satisfaction, mental health, and well-being (e.g., [46]; [50]; [89]), making it a central component in their successful adaptation to university life. Considering this, it becomes clear that the development of SEC must continue into higher education, through intentional assessment and SEL interventions, not only as a skills-based intervention but also as a way of shaping need-supportive learning environments to promote overall well-being and prevent mental health problems.

Evidence consistently highlights the importance of SEC in promoting students’ well-being, with emotional intelligence identified as a key construct underlying this skill set (e.g., [19]; [20]; [23]; [56]; [75], [78]). By providing students with the tools to navigate complex interpersonal and academic challenges more effectively, emotional intelligence is moderately associated with students’ enhanced psychological well-being ([17]). Among the specific behaviors shaped by SEC are the willingness to seek support, the ability to communicate emotional needs, and the capacity to engage in healthy interpersonal dynamics ([30]; [77], [78], [75]). Conversely, when students lack these competencies, they may struggle to cope with the pressures of personal and academic life, which can increase their likelihood of engaging in risk-taking behaviors and experiencing poor academic performance ([22]; [33]). Seeking help, particularly, is a critical yet often neglected element of students’ adaptation to university life. While accessing support services can significantly buffer the impact of psychological distress, students experiencing high levels of strain are often less likely to seek professional help ([43]). Yet, help-seeking is a complex process shaped by both the availability of social support and individual perceptions. Together, prior literature suggests that promoting SEC in higher education is not only beneficial but essential for equipping students with personal resources needed to navigate the challenges of emerging adulthood and sustain long-term well-being.

In addition to the well-established relevance of higher education students’ SEC, research has also highlighted and discussed specific subgroups where the needs and expression of these competencies may differ, namely gender and academic fields ([23]). Accordingly, these variables constitute key areas of interest that will be briefly addressed in this introduction and considered in the present study.

#### 1.2.1. SEL Differences Across Gender

Research in higher education also reveals gender differences in students’ mental health, often showing that female students tend to report higher levels of mental health literacy, but that they also experience greater psychological distress (e.g., anxiety and depression symptoms) and lower overall well-being compared to their male peers. Although these gender differences may vary across cultural contexts ([40]; [43]), and be linked to biological and social factors (e.g., [39]), they are also associated with differences in coping strategies, emotional regulation, and other social and emotional skills (e.g., [113]). Female students tend to score higher in emotion-focused coping, emotional intelligence, and empathy (e.g., [42]; [44]), which may contribute to different ways of managing academic and emotional distress. Despite this, women still report more emotional distress, possibly due to heightened emotional awareness, societal norms, and gender-role expectations regarding emotional expression. In contrast, male students may underreport emotional difficulties or avoid seeking help more often ([2]).

#### 1.2.2. SEL in Different Higher Education Fields

##### Science, Technology, Engineering, and Mathematics Fields

On one hand, as technological innovation becomes central to society and the global economy, more students are drawn to Science, Technology, Engineering, and Mathematics fields (STEM; e.g., biology, chemistry, mathematics, computer science, data science, robotics) ([93]). However, this trend also reveals the distinct academic and emotional challenges that can compromise their persistence and success in higher education ([18]; [82]; [108]). One of the most common challenges faced by students in the early stages of their degree is a loss of interest and motivation, often linked to low grades and feelings of discouragement ([82]). This can gradually undermine their confidence in their academic abilities ([100]). STEM programs are frequently characterized by competitive and unsupportive cultures ([100]), where students may feel isolated or struggle to develop a sense of belonging. These factors may contribute to students’ increased stress, anxiety, and burnout ([47]; [95]) and reduce their engagement and investment in the learning process ([18]). Additionally, structural barriers such as the difficult transition from high school to college, weed-out classes, and intense course loads—including overloaded schedules, challenging lab work, and fast-paced instruction—also contribute to dropout risk and emotional distress ([100]). Moreover, women in STEM higher education continue to face significant obstacles, including stereotypes, gender bias, and limited access to mentoring ([12]). These factors contribute to unstable academic identities and a weak sense of belonging, which can affect their engagement and confidence, and act as a barrier to degree completion ([12]). Research also shows that academic support impacts male and female STEM students differently. While academic support tends to benefit male students, it appears to be less effective for female students. Also, male students gain more from autonomous (self-driven) motivation, whereas female students are more affected by controlled motivation, which is linked to external pressure and often leads to negative outcomes ([47]). Understanding and addressing the factors that influence persistence among all STEM students is essential for fostering inclusive and equitable educational environments. Together, these factors underscore the urgent need for institutions to support not only the academic success of STEM students but also their social and emotional development ([18]).

##### Humanities, Arts, and Social Sciences Fields

On the other hand, less literature has been focused on the Humanities, Arts, and Social Sciences (HASS; e.g., history, psychology, education, communication and media studies, law, economics, and visual arts) compared to STEM disciplines ([54]). This discrepancy may be partly explained by a longstanding institutional undervaluing of these fields, often reflected in reduced funding. This lower prioritization is also reflected in structural decisions, including the suspension of specific courses or entire departments ([24]), thereby limiting both academic visibility and the advancement of field-specific research and interventions.

Moreover, distinct challenges apply to students in HASS fields. Unlike STEM students, HASS students often perceive the main difficulties not during their academic journey but rather in the uncertainty surrounding post-graduation employment prospects. Their primary sources of stress tend to revolve less around academic performance and success but more around navigating an ambiguous and unstable job market after graduation ([49]; [69]). This perspective introduces an additional layer of psychological and emotional strain, underscoring the importance of context-specific support mechanisms and the development of domain-relevant social and emotional competencies.

Academic motivation also appears to vary across academic fields. Students in HASS tend to report higher intrinsic motivation, whereas those in STEM are more likely to display extrinsic motivation and even demotivation ([65]). In parallel, significant disparities in mental health outcomes have been observed across fields of study. Evidence consistently shows that students in HASS are more likely to experience psychological distress than their peers in STEM fields ([61]; [70]). Collectively, these findings highlight the need for differentiated psychological and pedagogical support strategies, particularly in academic contexts where emotional vulnerability and future-oriented uncertainty are more pronounced.

### 1.3. Present Study

Given the identified gap in assessing and promoting students’ SEC in higher education contexts, our study aims to address this pressing need by adapting the *Social and Emotional Competence Assessment Battery for Adults—General Survey* (SECAB-A; [81]) for use with higher education students. The *Social and Emotional Competence Assessment Battery for Adults—Students Survey* (SECAB-A(S)) is a context-specific instrument that allows researchers and practitioners to capture the specific dynamics of students’ SEC in university environments. In this study, we intend to evaluate the psychometric properties of the SECAB-A(S), namely its structural, discriminant, and concurrent criterion validity, reliability, and multi-group invariance across gender and academic fields.

In line with the SEL framework and prior results with the SECAB-A General Survey ([81]), the following hypotheses were formulated regarding the expected factor structure of the SECAB-A(S):

**Hypothesis 1a** **(H1a).**
*The Intrapersonal Competence Questionnaire is expected to reveal a first-order two-factor structure (self-awareness and self-regulation).*


**Hypothesis 1b** **(H1b).**
*The Interpersonal Competence Questionnaire is anticipated to also present a first-order two-factor solution (positive relationship and conflict management).*


**Hypothesis 1c** **(H1c).**
*The Responsible Decision-Making Competence Questionnaire is expected to demonstrate a unidimensional structure, targeting the construct of responsible decision-making.*


To test the construct validity of the SECAB-A(S), we examined its discriminant validity by comparing it with an external measure of affective relationship satisfaction. While SEC may contribute to positive social relationships, the SECAB-A(S) is designed to measure broader social and emotional skills that go beyond the scope of romantic or affective satisfaction. Demonstrating discriminant validity ensures that the SECAB-A(S) captures distinct dimensions of competence, rather than overlapping significantly with related but conceptually different constructs. The following hypothesis was established regarding discriminant validity:

**Hypothesis 2** **(H2).**
*Small and positive intercorrelations are expected between the SECAB-A(S) scales and satisfaction with affective relationships.*


We also expect to find concurrent criterion validity between the SECAB-A(S) and students’ personal and academic well-being dimensions. Following prior literature, we expect a positive association between students’ social and emotional skills and their well-being. The following hypothesis was defined:

**Hypothesis 3** **(H3).**
*Moderate to large positive intercorrelations between SECAB-A(S) scales and personal and academic well-being dimensions are expected.*


Additionally, we examined the reliability and multi-group measurement invariance (configural, metric, scalar, and strict) of the SECAB-A(S) across gender and academic fields, as these constitute relevant sociodemographic subgroups identified in prior literature on higher education students. We expect to attain good internal consistency of the SECAB-A(S) scales and to establish structural equivalence across groups (e.g., gender and academic fields), reducing measurement bias and allowing cross-group comparisons. Contingent upon establishing the SECAB-A(S) adequacy, we intend to investigate potential differences in students’ SEC across gender and academic fields, as prior research has pointed out expected differences. We also intend to explore the direct impact of students’ SEC on their personal and academic well-being (controlling for gender, age, and academic fields). Following the prior literature, we established the following research question and hypothesis:

**Research Question 1** **(Q1).**
*Do higher education students perceive SEC differently based on their gender and their academic field?*


**Hypothesis 4** **(H4).**
*Students’ SEC will positively predict students’ personal and academic well-being.*


## 2. Materials and Methods

### 2.1. Participants

A total of 767 higher education students (62.8% female, *M* = 22.88 years, *SD* = 7.30) enrolled in the study. Most participants were Portuguese (92.8%), did not have any special academic status (91.5%), and studied in the same geographical area of residence (64.0%). Although a non-probability sampling method was used, our sample included students from all Portuguese counties and education and training fields (with over 100 degree programs represented), ensuring national representation. Although most participants were undergraduate students (68.7%), our sample also included master’s and PhD students. Table 1 depicts the sociodemographic characterization of the sample in comparison to the Portuguese population of higher education students.

### 2.2. Measures

The data were collected through self-report questionnaires to assess social–emotional competence, and personal and academic well-being. Socio-demographic data were also collected (gender, age, nationality, course and university, special student status, and place of residence relative to permanent home).

#### 2.2.1. Social and Emotional Competence

Students’ SEC was assessed through the *Social and Emotional Competence Assessment Battery for Adults—Students Survey* (SECAB-A(S)). As SEC is context-dependent, the Student Survey was adapted in the context of this study from the SECAB-A General Survey ([81]) to better assess the use of social and emotional skills in higher education contexts. The SECAB-A(S) is composed of three independent questionnaires with a total of 37 items that assess self-awareness (7 items, ω = 0.81), self-regulation (8 items, ω = 0.84), positive relationship skills (8 items, ω = 0.77), conflict management skills (8 items, ω = 0.73), and responsible decision-making (6 items, ω = 0.78). Items (e.g., “During stressful moments at university, I am able to stay calm.”) were rated on a 10-point scale (from 1—Never to 10—Always).

#### 2.2.2. Satisfaction with Affective Relationships

Students’ satisfaction with their affective relationships was assessed with an adaptation of the 3-item *Kansas Marital Satisfaction Scale* (KMSS; [99]; Portuguese version: [4]) (ω = 0.99). Items (e.g., “How satisfied are you with your relationship?”) were rated on a 7-point scale (from 1—*Extremely dissatisfied* to 7—*Extremely satisfied*).

#### 2.2.3. Personal Well-Being

We used the *Mental Health Continuum—Short Form* (MHC-SF, [53]; Portuguese version: [66]) to measure students’ personal well-being. The questionnaire includes 14 items focusing on feelings of emotional (3 items, ω = 0.86), psychological (6 items, ω = 0.86), and social (5 items, ω = 0.81) well-being. Items (e.g., “how often have you felt happy?”) were rated considering the frequency of the described feeling in the previous month on a 6-point scale (from 0—Never to 5—Every day).

#### 2.2.4. Academic Well-Being

Academic well-being was measured with the 9-item version of the *Utrecht Work Engagement Scale for students* (UWES-S, [96]; Portuguese version: [97]). The UWES-S measures feelings of vigor (3 items, ω = 0.91), dedication (3 items, ω = 0.89), and absorption (3 items, ω = 0.82). Students rated, on a 7-point Likert scale (from 0—Never to 6—Every day), how often they had experienced those feelings (e.g., “I am enthusiastic about my studies”).

### 2.3. Procedures

#### 2.3.1. Data Collection

Prior to data collection, the Ethics and Deontology Committee of the Faculty of Psychology, University of Lisbon, granted approval of the study (protocol code *Ata n°9/2023* and *Ata n°4/2024*). Measures and the socio-demographic questions were uploaded as an online survey using the *Qualtrics* platform (average response time: 15 min). The anonymous survey link, along with information regarding the study’s purpose, was launched via email to universities as well as departments, associations, and student unions, asking for their collaboration in the dissemination of the survey through mailing lists. We also launched the survey on social networks and student groups and through the researchers’ direct contact networks. This method enabled us to reach higher-education students from all the Portuguese counties and represent all education and training fields in compliance with the Directorate General for Higher Education’s (DGES) classification (Table 1). The only eligibility criterion was that the participants had to be students currently enrolled in a Portuguese higher education institution. Participants were self-selected based on voluntary enrolment, and informed consent was guaranteed before partaking. Participation was anonymous, and data confidentiality was guaranteed. No compensation was offered to the participants. Data were collected in two cross-sectional waves between May 2024 and May 2025. In the first wave (*n* = 538; May 2024–March 2025), we only collected data regarding SEC to test the factorial structure of the SECAB-A(S). In the second wave (*n* = 229; April–May 2025), we applied all the measures to confirm the factor structure of the SECAB-A(S), test the invariance of the measure and its discriminant validity against external measures (KMSS), and test the relationship of SEC and students’ well-being (MHC-SF and UWES-S). As the data were collected online, in the event of missing values, the software prompted participants to complete their responses prior to submission, leading to no missing data. To ensure online data quality and validity, we applied a data validation protocol with the following criteria: consistency of response; use of text entry boxes to facilitate the detection of random answers, spam, or the use of autofill software; tracking for multiple response submissions; and a minimum threshold of 5 min response completion time ([7]; [27]). A statement promoting honesty was added in the survey instructions, and an honesty question asking how many questions were answered truthfully was included at the end of the survey to mitigate social desirability bias and contribute to response validity screening ([59]). Responses that did not meet the data validation protocol criteria were deleted.

#### 2.3.2. Data Analysis

Data analyses were performed using IBM SPSS Statistics v.29 and the R environment software (version R 4.2.0; [87]). For sample size definition, we ensured a participant-to-parameter ratio of 10:1 ([55]) and computed an a priori power analysis for the CFA model (power = 0.80, *p* = 0.05, and RMSEA < 0.05; [73]), which indicated a minimum of 182 participants to test the structural model. We performed a data diagnosis verifying assumptions of adequate correlation between variables (Bartlett test with *p* < 0.05, KMO > 0.05, and VIF < 5; [51]; [71]) and normal distribution of the data (*Q-Q* plot analysis with |z| > 3; [55]).

To test the structural factor model of the measure, we computed a confirmatory factor analysis (CFA) to test the fit of the three independent models proposed by the authors for the SECAB-A General Survey ([81]). Models were compared with additional solutions to determine which of the alternatives best fit the data. Model fit was evaluated through the following fit indices: Chi-squared test (χ^2^), chi-squared/degrees of freedom (χ^2^/df), the comparative fit index (CFI), the Tucker–Lewis index (TLI), the standardized root mean square residual (SRMR), the root mean square error of approximation (RMSEA) with a 90% confidence interval, Akaike information criteria (AIC), and Bayesian information criteria (BIC). An adequate fit was considered for a χ^2^/df value below 5 ([5]), CFI and TLI values close to 0.90 or above ([9]; [10]), and SRMR and RMSEA values below 0.08 ([5]; [48]). As for model comparison, smaller AIC and BIC values (thus suggesting a more parsimonious solution; [5]; [15]) and the chi-square difference test against alternative models ([13]) were considered. Model specification analyses were performed, and modification indices (MI; cutoff of >15) were included in the models reproducing the adjustments proposed by the authors for the SECAB-A General Survey ([81]) or when theoretically supported.

Additional scale diagnosis was performed to evaluate reliability and discriminant and criterion validity. For reliability, coefficient omega was computed and considered good for scores equal to or above 0.70 ([25]). Discriminant validity was tested against the external measure of satisfaction with affective relationships. This variable was chosen to assess discriminant validity since, following prior literature, it is expected to be positively related but distinct from SEC. Evidence of discriminant validity occurred for small correlations ([55]). Concurrent criterion validity was tested against indicators of students’ personal and academic well-being and occurred for moderate to large intercorrelations. Correlations are considered small, moderate, and large for values around 0.10, 0.30, and 0.50, respectively ([21]). We tested multi-group invariance of the SECAB-A(S) across gender and academic fields. Four increasingly constrained models were tested: configural, metric, scalar, and strict. Invariance was assessed based on established criteria, with differences in fit indices interpreted as evidence of invariance when ΔCFI ≤ 0.010 and ΔRMSEA ≤ 0.015 ([86]).

Considering the differences between academic fields (cf. Introduction), in the context of this study, we grouped academic fields into two clusters. Our decision was informed by the [85] ([85]) classification for education and training fields, prior scientific literature on higher education, and the nature of each discipline. That said, the first cluster, named Science, Technology, Engineering, and Mathematics (STEM), comprises the following disciplines: agriculture, forestry, fisheries, and veterinary sciences; natural sciences, mathematics, and statistics; engineering, manufacturing, and construction; and information and communication technologies. The second group, designated Humanities, Arts, Social Sciences and Health (HASS-H), covers education; arts and humanities; social sciences, journalism, and information; business sciences, administration, and law; health and social protection; and services. This classification mirrors common classifications in educational research and supports theoretically grounded comparisons between technical-scientific and human-social academic fields.

Lastly, associations between SEC and sociodemographic variables and well-being were tested with Pearson correlations. Mean differences were computed between groups for gender and academic field using independent samples *t*-test. Effect sizes were estimated using Cohen’s *d*. Effect sizes were considered small, moderate, and large for values around 0.20, 0.50, and 0.80, respectively ([21]). We computed multiple hierarchical regression analyses to test whether SEC predicted students’ personal and academic well-being, controlling for sociodemographic variables (age, gender, and academic field). Assumptions for applying regression models were verified through the graphical analysis of the studentized residuals, the Durbin-Watson statistic (≈2), and VIF (<5). Significant effects were considered for *p* < 0.05 and whenever the 95% CI did not include 0.

## 3. Results

### 3.1. Data Diagnosis

Data diagnosis revealed adequate correlations between variables for each questionnaire (*Intrapersonal Competence Questionnaire*: *χ*^2^(105) = 4385.62, *p* < 0.001, overall KMO = 0.90, item KMOs > 0.86; *Interpersonal Competence Questionnaire*: *χ*^2^(120) = 3139.19, *p* < 0.001, overall KMO = 0.87, item KMOs > 0.74; *Responsible Decision*-*Making Competence Questionnaire*: *χ*^2^(15) = 1186.31, *p* < 0.001, overall KMO = 0.80, item KMOs > 0.78). No evidence of multicollinearity was observed (VIF range: 1.09 to 2.27). Analyses of *Q-Q* plots suggested a tendency towards normal distribution of the data, with most data points being clustered around 0 and not surpassing 1.5 standard deviations. Thus, maximum likelihood estimation was used for CFA models.

### 3.2. Confirmatory Factor Analysis

Goodness-of-fit indices for the models under study and alternative solutions integrating MI are illustrated in Table 2. Model fits were better for the structures replicating the SECAB-A General Survey ([81]). Figure 1, Figure 2 and Figure 3 present the final factor structures of the three SECAB-A(S) questionnaires.

#### 3.2.1. Intrapersonal Competence Questionnaire

Initial CFA suggested that, when comparing the alternative models, the first-order structure with two factors (Model B) best fitted the data (*χ*^2^(89) = 636.11, *p* < 0.001, *χ*^2^/*df* = 7.15, CFI = 0.82, TLI = 0.79, SRMR = 0.08, RMSEA = 0.09, 90% CI [0.09, 0.10]). However, the model still had a poor fit. Examination of MI informed adjustments to improve the models’ fit. Items 02 and 06 (MI = 102.64) and Items 08 and 10 (MI = 171.96) were forced to covary, replicating the MI included in the SECAB-A General Survey. Additionally, Item 03 was forced to covary with Item 01 (MI = 32.35), Item 02 (MI = 86.67), and Item 06 (MI = 22.16). These four items relate to emotional recognition and regulation. Items 13 and 14 (both related to the ability to take different perspectives) should also display error covariances (MI = 39.09). CFA of the re-specified models suggested that Model B had an adequate fit (*χ*^2^(83) = 333.20, *p* < 0.001, *χ*^2^/*df* = 4.01, CFI = 0.92, TLI = 0.90, SRMR = 0.06, RMSEA = 0.06, 90% CI [0.06, 0.07]). The re-specified Model B also showed substantially improved fit over Model A. Model C does not provide a statistically significant improvement over Model B.

#### 3.2.2. Interpersonal Competence Questionnaire

Initial CFA revealed that, while the first-order structure with two factors (Model B) best fitted the data in comparison to the alternative models, goodness-of-fit statistics did not support acceptability of the model (*χ*^2^(89) = 460.12, *p* < 0.001, *χ*^2^/*df* = 5.17, CFI = 0.83, TLI = 0.80, SRMR = 0.06, RMSEA = 0.09, 90% CI [0.08, 0.09]). Examination of MI led to the following adjustments to improve the models’ fit: following the MI applied in the SECAB-A General Survey, Items 09 and 10 were forced to covariate (MI = 84.21); additionally, Items 04 and 05 (MI = 30.75; both resorting to open communication), Items 11 and 12 (MI = 45.35; both focusing on respectful social interactions), Items 14 and 15 (MI = 39.95; reflecting active listening), and Item 03 with Items 02 (MI = 35.18) and 11 (MI = 21.57) (linked to empathy and social awareness) were forced to display error covariances. After integrating MI, CFA evidenced that the re-specified Model B adequately fitted the data (*χ*^2^(83) = 272.84, *p* < 0.001, *χ*^2^/*df* = 3.29, CFI = 0.92, TLI = 0.90, SRMR = 0.05, RMSEA = 0.06, 90% CI [0.05, 0.07]). Model A had a poor fit to the data, and Model C offered no practical advantage, as its adequacy was statistically equal to Model B, but with increased complexity.

#### 3.2.3. Responsible Decision-Making Competence Questionnaire

Initial CFA revealed a poor fit of the unidimensional model (*χ*^2^(89) = 636.11, *p* < 0.001, *χ*^2^/*df* = 7.15, CFI = 0.82, TLI = 0.79, SRMR = 0.08, RMSEA = 0.09, 90% CI [0.09, 0.10]). Examination of MI indicated that, following the MI included in the SECAB-A General Survey, Items 01 and 02 should display error covariances (MI = 95.20). The new CFA, integrating MI, revealed an adequate fit of the re-specified model (*χ*^2^(8) = 24.92, *p* < 0.001, *χ*^2^/*df* = 3.12, CFI = 0.98, TLI = 0.96, SRMR = 0.03, RMSEA = 0.05, 90% CI [0.03, 0.07]).

### 3.3. Factorial Invariance Analysis

Table 3 includes the data related to measurement invariance across gender (female vs. male) and academic field (STEM vs. HASS-H). Factorial invariance across groups was computed to test whether the latent structure of the fitted models remained similar when comparing female (*n* = 461) and male (*n* = 265) students and STEM (*n* = 513) and HASS-H (*n* = 219) students.

Multi-group measurement invariance of the *Intrapersonal Competence Questionnaire* was examined. Across genders, metric invariance was supported. Although full scalar invariance was not achieved (ΔCFI = 0.014), a partial scalar model showed acceptable fit (ΔCFI = 0.009; ΔRMSEA = 0.001). Residual invariance was also supported (ΔCFI = 0.003). Across academic fields, configural, metric, and scalar invariance were supported. Full residual invariance was not attained (ΔCFI = 0.017), but a partial model met the criteria (ΔCFI = 0.010; ΔRMSEA = 0.001).

For the *Interpersonal Competence Questionnaire*, metric invariance was supported across gender. Full scalar invariance was not achieved (ΔCFI = 0.019), but partial scalar invariance met the established criteria (ΔCFI = 0.008; ΔRMSEA = 0.000). Partial residual invariance was also supported (ΔCFI = 0.007). Across academic fields, configural, metric, and scalar invariance were supported. Full residual invariance was not supported (ΔCFI = 0.015), but a partial model demonstrated acceptable fit (ΔCFI = 0.009; ΔRMSEA = 0.000).

Full measurement invariance (configural, metric, scalar, and residual) was established for the *Responsible Decision-Making Competence Questionnaire* across both gender and academic field, with all model comparisons falling within acceptable thresholds.

### 3.4. Reliability, Discriminant and Criterion Validity, and Correlation Analysis

Coefficient omegas were adequate, and correlations between scales were moderate to large (Table 4). As anticipated, intercorrelations between the SECAB-A(S) scales and students’ satisfaction with affective relations were small, suggesting discriminant validity (Table 5). Intercorrelations between the SECAB-A(S) scales and students’ personal and academic well-being dimensions are depicted in Table 6. Self-awareness and personal and academic well-being dimensions had generally small, positive, and significant correlations. Self-regulation presented generally large, positive, and significant intercorrelations with personal and academic well-being dimensions. For interpersonal and responsible decision-making skills, we found generally moderate, positive, and significant intercorrelations with personal and academic well-being dimensions. Associations between age and self-regulation and positive relationship skills were small, positive, and statistically significant. The remaining correlations between the SECAB-A(S) scales and sociodemographic indicators (age, gender, and academic field) were extremely small, with variables being barely related (as they were below the threshold of 0.10).

### 3.5. Group Differences

Male students perceived higher self-regulation competencies than female students (*t* = −2.314, *p* = 0.021, *d* = −0.18) (Table 7). Additionally, students in HASS-H perceived higher intrapersonal skills [self-awareness (*t* = 2.632, *p* = 0.004, *d* = 0.21) and self-regulation (*t* = 1.911, *p* = 0.028, *d* = 0.16)] than students in STEM (Table 8). No differences were found for interpersonal skills and responsible decision-making across gender or academic field.

### 3.6. Regression Analysis

Two hierarchical models were considered: Model 1 included sociodemographic indicators as control variables (i.e., age, gender, and academic field). Model 2 added the SEC variables as predictors of personal and academic well-being dimensions.

#### 3.6.1. Personal Well-Being

##### Emotional Well-Being

Model 2 proved to be statistically significant [*F*(8, 217) = 13.041, *p* < 0.001, *R*^2^ = 0.325, ∆*R*^2^ = 0.32], explaining around 33% of the variance of emotional well-being. Analysis of the regression coefficients and their statistical significance evidenced that, of the predictors considered, both self-awareness (*β* = −0.18, *t* = −2.24, *p* = 0.026, 95% CI [−0.26, −0.02]) and self-regulation (*β* = 0.68, *t* = 7.80, *p* < 0.001, 95% CI [−0.34, 0.57]) were significant predictors of students’ emotional well-being. Age was also a significant predictor of students’ emotional well-being (*β* = −0.13, *t* = −2.24, *p* = 0.026, 95% CI [−0.03, −0.01]).

##### Psychological Well-Being

Model 2 proved to be statistically significant [*F*(8, 217) = 22.937, *p* < 0.001, *R*^2^ = 0.458, ∆*R*^2^ = 0.43], explaining around 46% of the variance of social well-being. The individual predictors were examined further and indicated that only self-regulation (*β* = 0.67, *t* = 8.52, *p* < 0.001, 95% CI [0.37, 0.59]) was a significant predictor of students’ psychological well-being.

##### Social Well-Being

Model 2 proved to be statistically significant [*F*(8, 217) = 8.014, *p* < 0.001, *R*^2^ = 0.228, ∆*R^2^* = 0.19], explaining around 23% of the variance of social well-being. Analysis of the regression coefficients and their statistical significance evidenced that both self-awareness (*β* = −0.22, *t* = −2.59, *p* = 0.010, 95% CI [−0.32, −0.04]) and self-regulation (*β* = 0.48, *t* = 5.08, *p* < 0.001, 95% CI [0.21, 0.48]) were significant predictors of students’ social well-being. Gender was also a significant predictor of students’ social well-being (*β* = 0.15, *t* = 2.40, *p* = 0.017, 95% CI [0.06, 0.63]).

#### 3.6.2. Academic Well-Being

##### Vigor

Model 2 proved to be statistically significant [*F*(8, 216) = 16.566, *p* < 0.001, *R*^2^ = 0.380, ∆*R*^2^ = 0.29], explaining around 38% of the variance of feelings of vigor. Analysis of the regression coefficients and their statistical significance evidenced that, of the predictors considered, both self-regulation (*β* = 0.67, *t* = 7.94, *p* < 0.001, 95% CI [0.56, 0.93]) and age (*β* = 0.14, *t* = 2.44, *p* = 0.015, 95% CI [0.01, 0.05]) were significant predictors of students’ vigor.

##### Dedication

Model 2 proved to be statistically significant [*F*(8, 216) = 12.453, *p* < 0.001, *R*^2^ = 0.316, ∆*R*^2^ = 0.27], explaining around 32% of the variance of students’ dedication. The individual predictors were examined further and indicated that both self-awareness (*β* = −0.18, *t* = −2.185, *p* = 0.030, 95% CI [−0.34, −0.02]) and self-regulation (*β* = 0.52, *t* = 5.85, *p* < 0.001, 95% CI [0.34, 0.67]) were significant predictors of students’ dedication.

##### Absorption

Model 2 proved to be statistically significant [*F*(8, 216) = 15.173, *p* < 0.001, *R*^2^ = 0.360, ∆*R*^2^ = 0.31], explaining around 36% of the variance of feelings of absorption. The individual predictors were examined further and indicated that self-awareness (*β* = −0.27, *t* = −3.467, *p* < 0.001, 95% CI [−0.47, −0.13]) and self-regulation (*β* = 0.63, *t* = 7.432, *p* < 0.001, 95% CI [0.44, 0.77]) were both significant predictors of students’ feelings of absorption.

## 4. Discussion

### 4.1. Interpretation of Key Findings

In recent years, the relevance of SEC has gained renewed attention, particularly within higher education and workplace contexts. As higher education students or soon thereafter as employees, individuals are increasingly required to navigate complex, uncertain, and rapidly evolving (work) environments ([23]; [30]; [60]). Given that robust evidence links SEC to improved health and well-being, performance, interpersonal relations, and leadership skills across diverse fields (e.g., educational, business, leadership, and health care), organizations are progressively seeking a socially and emotionally competent workforce ([81]). However, a clear gap remains in the assessment and promotion of SEC within higher education settings ([23]; [88]). In this study, we sought to address this gap by studying the psychometric properties of the SECAB-A(S), a context-specific adaptation of the SECAB-A General Survey ([81]) designed to capture students’ use of social and emotional skills in higher education contexts. Moreover, building on prior literature, we also aimed to study the differences in higher education students’ SEC across gender and academic fields (STEM vs. HASS-H) and to further explore the associations between students’ SEC and their personal and academic well-being.

#### 4.1.1. Psychometric Properties of the SECAB-A(S)

As anticipated, our findings provided evidence for the adequacy, validity, and reliability of the SECAB-A(S) in a sample of Portuguese higher education students. The CFA, replicating the factorial structure of the SECAB-A General Survey, yielded adequate goodness-of-fit indices while retaining the modification indices imposed in the original study. Accordingly, hypotheses H1a, H1b, and H1c were supported. We also found moderate to large positive intercorrelations between the SECAB-A(S) scales. These findings reinforce the structural validity of the measure. Furthermore, sustaining H2, we found small positive correlations between the SECAB-A(S) scales and students’ satisfaction with affective relations. This finding supports the measure’s discriminant validity, indicating that the SECAB-A(S) assesses a construct that is related to, but distinct from, satisfaction with affective relations. Together, these findings substantiate the instrument’s conceptual coherence and provide strong evidence for its construct validity. Additionally, the positive and significant moderate-to-large associations between the SECAB-A(S) scales and indicators of personal and academic well-being provide evidence of concurrent criterion validity (except for the self-awareness scale), thus partially supporting H3. Good coefficient omegas supported the reliability of the measure.

Multi-group measurement invariance analyses across the three questionnaires confirmed that the minimum criteria for metric invariance were met, ensuring equivalent item loadings across groups and supporting valid mean-level comparisons ([15]; [102]). We found partial residual invariance across gender and academic fields for the *Intrapersonal Competence Questionnaire* and the *Interpersonal Competence Questionnaire*. Full residual invariance was established for the *Responsible Decision-Making Competence Questionnaire* across gender and academic fields. Overall, these results demonstrate adequate cross-group equivalence, particularly at the metric and scalar levels, reinforcing the robustness of the SECAB-A(S) across relevant sociodemographic subgroups.

#### 4.1.2. Differences in Higher Education Students’ SEC by Gender and Academic Field

Regarding Q1, we found extremely small intercorrelations between the SECAB-A(S) scales and gender, as well as small, positive, and statistically significant intercorrelations between intrapersonal skills and age, consistent with prior evidence from the original study of the SECAB-A General Survey. We also found small, positive, and statistically significant intercorrelations between positive relationship skills and age. These findings suggest that the perceived SEC of higher education students is not associated with gender but appears to develop with age. This supports the argument that, with the growing centrality of SEC in education, gender differences tend to be attenuated and that SEC can be learned and developed over time ([68]; [74]).

By testing gender and academic field invariance, our findings offer new insights into group-level differences in higher education students’ SEC. We found gender differences in self-regulation, with male students reporting higher levels of self-regulation skills than female students, although the effect size was small (*d* = 0.18). This result contrasts with previous literature on SEC in youth, which typically reports that girls exhibit higher levels of emotional, behavioral, and academic self-regulation ([30]; [39]; [67]). However, findings across the literature are not unanimous (e.g., [91]). Although research in higher education is limited, some studies have reported gender differences in self-regulation favoring male students ([52]). Importantly, whereas prior literature often focuses on specific skills of self-regulation, such as emotional regulation, the SECAB-A(S) captures a broader range of self-regulatory specific skills consistent with the SEL theoretical framework (e.g., emotional and behavioral regulation, goal setting and achieving, self-efficacy, adaptability, optimism, and organizational skills). Within this broader framework, we found support in prior literature showing that male students tend to report higher levels of self-efficacy and self-esteem, whereas female students may be more self-critical and more likely to underestimate their performance (e.g., [39]; [58]; [83]), potentially influencing their responses to the questionnaire. We did not find significant gender differences for self-awareness, interpersonal, or responsible decision-making skills.

Our findings also indicate that students from Humanities, Arts, Social Sciences, and Health fields reported significantly higher intrapersonal competences (i.e., self-awareness and self-regulation) when compared to their peers in Science, Technology, Engineering, and Mathematics fields. Although the effect sizes were small, these differences tend to align with previous literature that describes the challenges faced by STEM students, particularly concerning the development and expression of emotional and social skills ([3]; [100]). A possible explanation lies in the distinctive nature of academic disciplines, as STEM and HASS-H fields may differentially foster or inhibit the development and expression of different social and emotional skills. Particularly, HASS-H disciplines—due to their highly social and reflective characteristics—may place greater emphasis on relational skills and introspection, thereby enhancing the relevance and use of intrapersonal skills. Conversely, STEM curricula tend to prioritize technical, procedural, and analytical skills, potentially hindering the development and expression of SEC in academic contexts ([23]). However, there is a dearth of existing work assessing SEC in higher education, particularly comparing different academic fields ([3]), so further investigation is needed to draw sustained interpretations of our data.

In sum, and answering Q1, our findings suggest that perceived SEC among higher education students seems to vary as a function of gender and academic field. Nonetheless, given the self-report nature of the SECAB-A(S), our findings for group comparisons should be interpreted with caution, as responses may be influenced by gender-role expectations and social desirability bias ([39]). Future research should therefore adopt multi-method assessments (e.g., behavioral, observational, or performance-based measures) to more accurately capture the complexity of SEC differences across gender and academic fields. This would enhance understanding of whether such differences reflect actual behavioral patterns, metacognitive processes, or socially constructed self-perceptions.

#### 4.1.3. Associations Between Students’ SEC and Well-Being Outcomes

Lastly, regression analyses did not fully support H4. Our results revealed that intrapersonal competencies significantly predicted students’ personal and academic well-being. As expected, self-regulation emerged as a consistent positive predictor across all well-being dimensions. This adds to prior evidence linking regulatory skills to adaptive coping, academic engagement, and psychological functioning (e.g., [45]), suggesting that self-regulation skills might help students to directly adapt to life challenges and maintain and increase their personal and academic well-being.

In contrast, self-awareness was found to negatively predict emotional and social well-being, as well as dedication and absorption. Although initially unexpected, this result is supported by prior literature. While research shows that self-awareness is beneficial for self-regulation (e.g., [45]), it has not consistently shown direct benefits for mental health ([101]). Conversely, some empirical studies have highlighted associations between heightened self-awareness and obsessive thinking, psychological distress, and decreased happiness ([101]). These results may reflect the “paradox of self-absorption”, whereby heightened self-reflection, particularly in high-pressure academic contexts, can increase ruminative thought and hinder emotional detachment or the experience of flow, especially during learning activities that require deep absorption ([101]).

Taken together, these findings suggest that, in academic settings, self-awareness, when not accompanied by effective self-regulation, may not only fail to promote positive outcomes but even prove detrimental by increasing vulnerability to distress and emotional strain. Simply becoming aware of one’s thoughts, emotions, or needs can amplify internal focus, which, under conditions of high pressure or limited coping resources, may lead to rumination, self-criticism, or paralysis rather than constructive action. This is consistent with research suggesting that self-focused attention, particularly in stressful situations, can exacerbate psychological distress when it is not paired with regulation strategies that allow individuals to redirect attention or manage affective responses ([72]).

This underscores the importance of a dynamic interplay between metacognition and behavioral regulation, pointing to the need for integrative interventions that go beyond awareness-building to actively foster self-regulatory strategies. While the ability to recognize emotions, internal states, and needs (i.e., self-awareness) constitutes a foundational step in the adaptation process, our findings indicate that for students to experience greater well-being, self-awareness must translate into action—specifically, coupling self-awareness with self-regulation skills that enable them to manage emotions and behaviors and respond effectively to their needs. In this light, self-awareness alone may amplify internal dialog without necessarily promoting resolution or change, particularly under stress.

In contrast, self-regulation (from an agentic perspective) reflects the capacity to respond adaptively to such awareness through goal setting, emotional modulation, and behavioral adjustment. These competences are crucial for sustaining well-being in demanding academic environments ([23]). Therefore, while self-awareness can foster valuable introspection and insight, its impact on well-being is likely contingent upon contextual and personal factors, such as stress levels, coping strategies, and the ability to redirect attention adaptively. Our findings reinforce the central protective role of self-regulation, echoing prior literature (e.g., [29]), and highlight the need for educational approaches that support not just awareness but also the practical application of self-regulatory strategies.

Although this was not the central aim of our study, we also found gender and age differences in well-being outcomes. Male students expressed higher levels of social well-being than females, echoing prior findings ([83]). Additionally, younger students reported greater emotional well-being, whereas older students expressed higher levels of vigor. These age-related patterns mirror prior empirical evidence. Personal well-being tends to decrease with aging, possibly due to the cumulative effect of increasing responsibilities, life challenges, and personal experiences ([84]). In contrast, vigor refers to the ability to maintain high levels of energy and mental resilience while working, the willingness to invest effort in one’s work, and persistence even in the face of difficulties. The higher levels of vigor reported by older students may reflect a greater familiarity with the academic setting and its demands, a stronger sense of autonomy, and increased academic self-efficacy developed over time in higher education. Similar patterns of increased engagement among older students and employees have also been documented in previous research ([109]). In our findings, the academic field did not predict students’ personal or academic well-being.

### 4.2. Limitations

Our study is not without limitations. We used a convenience sample, which, while facilitating quick, straightforward, and cost-effective data collection, also presents constraints. Participants’ non-random self-selection based on availability and willingness may reduce generalizability, increasing selection bias and limited representativeness. To mitigate this limitation, we recruited students from various universities across the country to enhance diversity and minimize contextual bias. Also, data were collected over two academic years, which may introduce potential variability related to the school-year calendar. Participants’ responses may have been influenced by contextual factors such as exam periods, academic workload, or seasonal stressors, thus hindering the conclusions. This is particularly marked for the 2nd wave of data collection, which took place at the end of the school year. Although our sample was intentionally predominantly composed of Portuguese students, given that the instrument was specifically adapted to the Portuguese cultural and educational context, it would be valuable for future studies to examine the instrument’s performance in diverse higher education populations to establish its broader applicability. In particular, with the appropriate cultural and linguistic adaptations, it would be important to validate the instrument in other countries where Portuguese is the official language (e.g., Brazil). Additionally, the cross-sectional nature of our data limits causal inference. Although regression analyses revealed associations among variables, they do not allow for conclusions about directionality. Moreover, unmeasured variables (e.g., personality traits, prior social experiences, family or socioeconomic background) may have mediated or moderated the observed associations, warranting further research into additional factors impacting students’ SEC. We relied exclusively on self-report measures, which, although efficient and widely used in applied psychological research, are particularly susceptible to social desirability bias. Despite our efforts to increase response validity and control for social desirability bias, it cannot be overruled. The tendency to provide answers perceived as socially acceptable rather than fully accurate might inflate or distort some associations. As previously noted, this is particularly relevant in cross-group comparisons, which should be interpreted with caution given the potential influence of gender-role expectations and socially desirable response tendencies. Taken together, it would be important for future research to draw on longitudinal data and to integrate multi-method data collection approaches, including behavioral measures, third-party reports, and qualitative data, to strengthen causal interpretation and improve ecological validity of the findings. These approaches would provide a more enriched and in-depth understanding of participants’ perceptions and experiences. Lastly, the SECAB-A(S)’s response format employed a 10-point Likert-type scale without a true midpoint, which may have posed cognitive challenges for respondents due to the amplified scale length and limited their ability to express ambivalence or neutrality. The absence of a midpoint, while potentially encouraging decisiveness, may have inadvertently led to artificially polarized responses, thereby compromising data precision and interpretability. Nevertheless, this format addressed a limitation reported in prior studies that used a 5-point Likert-type scale, which yielded inflated means and reduced response discrimination ([80], [81], [79]). Future studies should explore the comparative adequacy and psychometric behavior of the SECAB-A(S) using a 7-point Likert-type scale, which may offer a more balanced compromise between sensitivity and cognitive load.

### 4.3. Study Impact

Our study advances important contributions to both research and practice by providing strong support for the validity, reliability, and adequacy of the SECAB-A(S) as a theoretically grounded, context-specific measure to assess higher education students’ social and emotional skills. Standing out from previously published measures, which often target younger populations, lack contextual adaptation for higher education settings, and are not grounded in comprehensive SEL frameworks ([81]), the SECAB-A(S) specifically addresses the unique developmental and contextual needs of higher education students. Thus, the SECAB-A(S) bridges a gap in the context of SEC assessment at this academic level. While SEL literature emphasizes the need to assess and promote SEC in higher education students, limited research has been conducted in this academic context ([30]). This may, in part, be motivated by the scarcity of available instruments that are both rigorously validated and widely adopted for use in higher education. The SECAB-A(S) thus emerges as a timely and novel resource, offering an assessment tool that is not only psychometrically sound but also tailored for the distinct challenges and environments encountered in higher education, differentiating it from existing generic measures.

The educational and practical relevance of the SECAB-A(S) should also be noted. This measure may contribute to methodologically robust studies, facilitating accurate monitoring of students’ SEC, supporting program evaluation, and enhancing cross-cultural comparative studies within higher education contexts. By presenting a valid, developmentally adjusted, and context-sensitive SEC assessment instrument, the SECAB-A(S) strengthens the design and evaluation of targeted interventions, thereby bridging the gap in holistic SEC assessment tools for this academic level. Furthermore, the SECAB-A(S) also stands out as a useful resource for practitioners, professionals in education, and psychologists. It can assist in the identification of competence gaps, provide clear guidance on priority intervention topics, and help to establish targeted action goals and strategies. This level of specificity provides stakeholders with a more precise foundation for designing institutional strategies aimed at enhancing student well-being, academic persistence, and social integration—dimensions widely recognized as critical for success in higher education.

Finally, the demonstrated associations between the SECAB-A(S) and well-being outcomes underscore its potential as both an evaluative and preventive tool for mental health. By facilitating the early identification of at-risk students and informing timely, targeted interventions, this measure offers a proactive means of aligning context-specific needs assessment with prevention initiatives and student support strategies in higher education settings. In sum, the SECAB-A(S) directly responds to the pressing call for a validated, contextually appropriate, and practically relevant tool to assess social and emotional skills in higher education, thereby offering both researchers and practitioners a valuable instrument with broad educational implications.

## 5. Conclusions

This study aimed to evaluate the psychometric properties of the SECAB-A(S), a theoretically grounded and context-specific self-report measure to assess higher education students’ social and emotional skills. Our findings supported the adequacy, validity, and reliability of the measure within a sample of Portuguese higher education students, endorsing the use of the SECAB-A(S) to assess social and emotional skills in academic environments. Results also supported cross-group equivalence regarding relevant sociodemographic subgroups (gender and academic field).

In addition to establishing the SECAB-A(S)’s psychometric properties, our results provided new insights into group-level differences in SEC among higher education students. Gender differences were observed in self-regulation, favoring male students. Also, students in HASS-H fields demonstrated higher intrapersonal competencies than those from STEM disciplines. These findings underscore the importance of needs assessment and targeted interventions tailored to the specific characteristics of student subgroups. Furthermore, SEC appears to develop progressively with age, underscoring the value of longitudinal support and strategic action to support continued SEL throughout students’ higher education journeys. Lastly, self-regulation emerged as a key predictor of students’ personal and academic well-being, while higher self-awareness was unexpectedly linked to lower well-being. This emphasizes the need for interventions designed to foster not only self-awareness (from a self-knowledge and identity development perspective) but also the practical application of self-regulation strategies (from an agentic perspective).

These findings represent a significant contribution in a context marked by 1) the need for socially and emotionally competent students and (future) workers and 2) the scarcity of validated, theoretically grounded, and context-sensitive tools for assessing SEC in higher education. The validated SECAB-A(S) thus offers a valuable resource for stakeholders to better understand, monitor, and foster students’ SEL, supporting evidence-based research and informed institutional practices. Ultimately, prioritizing SEC development may enhance both academic outcomes and future workforce readiness, making it a crucial area for continued research and institutional investment.

## Figures and Tables

**Figure 1 ejihpe-15-00162-f001:**
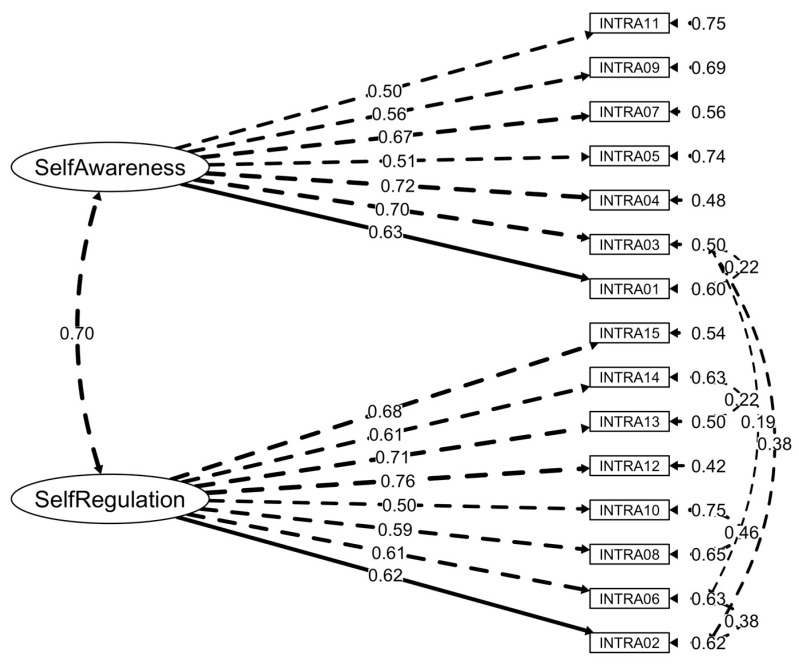
Factor structure and factor loadings of the SECAB-A(S) Intrapersonal Competence Questionnaire.

**Figure 2 ejihpe-15-00162-f002:**
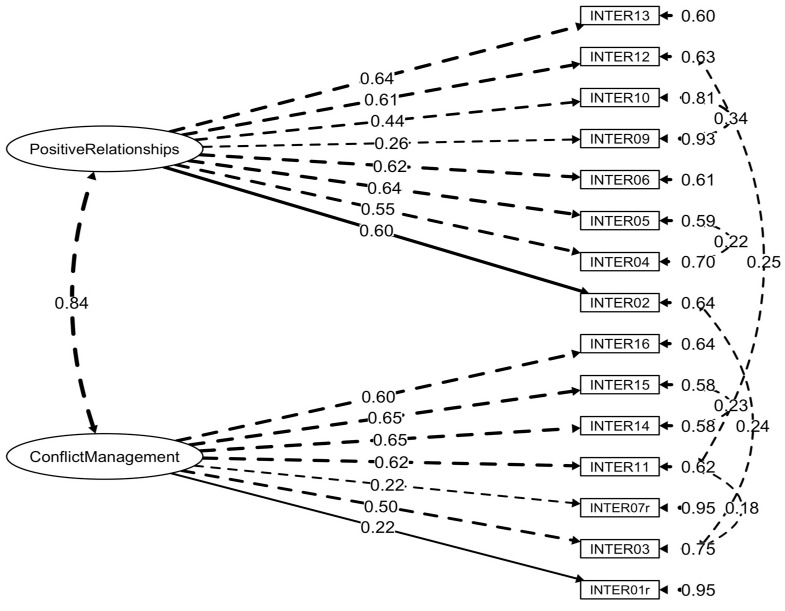
Factor structure and factor loadings of the SECAB-A(S) Interpersonal Competence Questionnaire.

**Figure 3 ejihpe-15-00162-f003:**
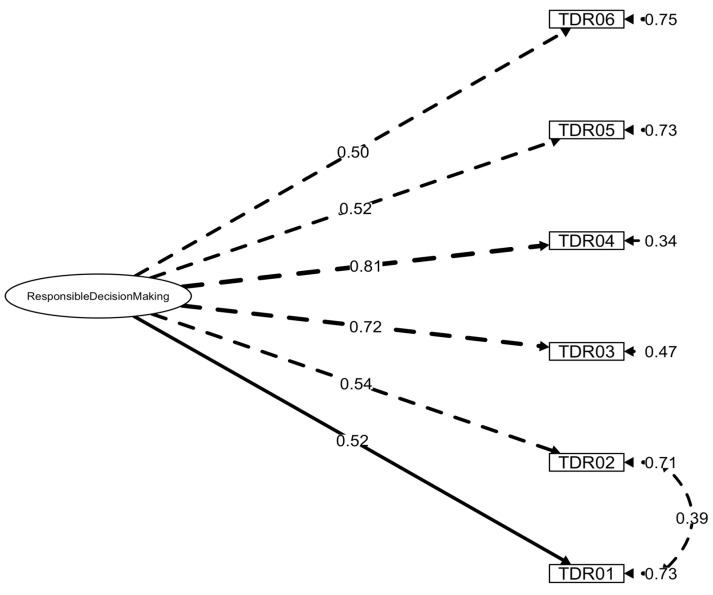
Factor structure and factor loadings of the SECAB-A(S) Responsible Decision-Making Competence Questionnaire.

**Table 1 ejihpe-15-00162-t001:** Participants’ sociodemographic characteristics (percentage of the most frequent category, mean, and standard deviation).

Variable	National Reference (*N* = 428,206)		Total Sample (*N* = 735)		
	%	μ	%	*M*	*SD*
Age		NA		22.88	7.30
*≤**18 years*	20.0		12.2		
*19–24 years*	62.9		71.1		
*25–29 years*	13.8		7.5		
*30+ years*	10.6		9.2		
Gender (Female)	53.7			62.8	
Nationality (Portuguese)	82.7			92.8	
Level of study					
*Undergraduate*	61.9		68.7		
*Master*	27.2		28.2		
*PhD*	5.8		2.6		
*Other*	4.9		0.5		
Education and training fields ^1^					
*Education*	3.8		3.0		
*Agriculture, forestry, fisheries and veterinary sciences*	2.3		5.2		
*Arts and humanities*	10.2		2.2		
*Natural sciences, mathematics and statistics*	5.7		28.1		
*Social sciences, journalism and information*	11.2		14.4		
*Engineering, manufacturing and construction*	19.8		35.4		
*Business sciences, administration and law*	21.9		2.7		
*Health and social protection*	15.7		6.4		
*Services*	5.8		1.2		
*Information and communication technologies (ICTs)*	3.4		1.4		
*Geral and non-specific*	0.1		0.3		
NUT II of study ^2^					
*North*	33.6			9.6	
*Center*	20.5			16.1	
*Lisbon Metropolitan Area*	37.3			65.4	
*Alentejo*	4.4			4.2	
*Algarve*	2.5			2.5	
*Autonomous Regions (Azores and Madeira)*	1.58			2.2	

NA = not available. ^1^ Data were categorized according to DGES classification and then grouped using the PORDATA cluster system; ^2^ Data were organized considering the Territorial Units for Statistical Purposes (NUT II). *Note*. The national reference data regarding the sociodemographic characteristics of the Portuguese population of higher education students were recovered from the latest data available on [34] ([34]) and [85] ([85]), reporting to the 2023/2024 school year.

**Table 2 ejihpe-15-00162-t002:** Goodness-of-fit statistics for the solutions of the Intrapersonal Competence Questionnaire, the Interpersonal Competence Questionnaire, and the Responsible Decision-Making Competence Questionnaire (n = 767).

	*χ^2^*	*df*	χ^2^/*df*	CFI	TLI	SRMR	RMSEAT	90% CI	AIC	BIC	df, ∆χ^2^	Model Comparison
*CFA for the re-specified models of the Intrapersonal Competence Questionnaire (15 items and modification indices)*												
Model A	552.42 ***	84	6.57	0.85	0.82	0.07	0.10	[0.09, 0.11]	44,055.85	44,221.45	_	_
Model B	333.20 ***	83	4.01	0.92	0.90	0.06	0.06	[0.06, 0.07]	43,758.43	43,928.63	*1*, 157.92 ***	Model A
Model C	329.19 ***	82	4.01	0.92	0.90	0.06	0.08	[0.07, 0.09]	43,760.43	43,935.23	*1*, 0.11	Model B
*CFA for the re-specified models of the Interpersonal Competence Questionnaire (16 items and modification indices)*												
Model A	366.48 ***	98	3.74	0.82	0.86	0.05	0.07	[0.06, 0.08]	48,089.13	48,263.93	_	_
Model B	272.84 ***	83	3.29	0.92	0.90	0.05	0.06	[0.05, 0.07]	45,066.76	45,236.95	*15*, 90.51 ***	Model A
Model C	269.56 ***	82	3.29	0.92	0.89	0.05	0.06	[0.06, 0.07]	45,068.76	45,243.55	*1*, −0.001	Model B
*CFA for the re-specified models of the Responsible Decision-Making Competence Questionnaire (6 items and modification indices)*												
Model A	24.92 ***	8	3.12	0.98	0.96	0.03	0.05	[0.03, 0.07]	17,261.08	17,320.88	_	_

*Note*. *χ*^2^ = Chi-Squared Test; Df = Degrees Of Freedom; CFI = Comparative Fit Index; TLI = Tucker–Lewis Index; SRMR = Standardized Root Mean Square Residual; RMSEA = Root Mean Square Error Of Approximation; AIC = Akaike Information Criteria; BIC = Bayesian Information Criteria. Model A (unidimensional structure), Model B (two first-order factors structure), Model C (one second-order factor structure). *** *p* < 0.001.

**Table 3 ejihpe-15-00162-t003:** Multigroup nested model comparisons.

	Overall Fit Indices				Comparative Fit Indices		
Invariance Models	*χ*^2^ (*df*)	CFI	TLI	RMSEA	Model Comparison	ΔCFI	ΔRMSEA
*Intrapersonal Competence Questionnaire*							
Gender groups							
*Configural*	580.85 (166)	0.92	0.90	0.06	_	_	_
*Metric*	595.35 (179)	0.92	0.90	0.07	Configural	0.000	0.003
*Scalar*	668.98 (192)	0.90	0.89	0.07	Metric	0.014	0.003
*Scalar_partial* ^1^	620.05 (191)	0.90	0.89	0.07	Metric	0.009	0.001
*Residual*	708.68 (208)	0.90	0.90	0.07	Scalar_partial	0.003	0.002
Academic field groups							
*Configural*	567.33 (166)	0.92	0.90	0.06	_	_	_
*Metric*	577.77 (179)	0.92	0.91	0.07	Configural	0.001	0.003
*Scalar*	621.40 (192)	0.92	0.91	0.06	Metric	0.007	0.000
*Residual*	709.95 (207)	0.90	0.90	0.06	Scalar	0.017	0.003
*Residual_partial* ^2^	675.94 (205)	0.90	0.90	0.07	Scalar	0.010	0.001
*Interpersonal Competence Questionnaire*							
Gender groups							
*Configural*	498.13 (166)	0.90	0.89	0.06	_	_	_
*Metric*	508.47 (179)	0.90	0.89	0.06	Configural	0.001	0.003
*Scalar*	577.20 (192)	0.88	0.86	0.07	Metric	0.019	0.003
*Scalar_partial* ^3^	544.91 (191)	0.90	0.89	0.06	Metric	0.008	0.000
*Residual*	580.76 (205)	0.86	0.86	0.07	Scalar_partial	0.007	0.000
Academic field groups							
*Configural*	445.57 (166)	0.91	0.89	0.08	_	_	_
*Metric*	461.95 (179)	0.91	0.90	0.07	Configural	0.001	0.002
*Scalar*	494.20 (192)	0.91	0.90	0.07	Metric	0.007	0.000
*Residual*	553.06 (207)	0.89	0.89	0.07	Scalar	0.015	0.002
*Residual_partial* ^4^	534.19 (206)	0.89	0.89	0.07	Scalar	0.009	0.000
*Responsible Decision-Making Competence Questionnaire*							
Gender groups							
*Configural*	51.71 (16)	0.97	0.95	0.06	_	_	_
*Metric*	65.93 (21)	0.97	0.95	0.07	Configural	0.008	0.002
*Scalar*	82.73 (26)	0.97	0.96	0.05	Metric	0.010	0.001
*Residual*	100.16 (32)	0.96	0.96	0.06	Scalar	0.010	0.001
Academic field groups							
*Configural*	50.91 (16)	0.98	0.96	0.07	_	_	_
*Metric*	56.08 (21)	0.98	0.97	0.06	Configural	0.000	0.010
*Scalar*	65.55 (26)	0.97	0.97	0.06	Metric	0.004	0.003
*Residual*	76.04 (32)	0.97	0.97	0.05	Scalar	0.004	0.003

^1^ freeing item 6; ^2^ freeing item 7 and item 9; ^3^ freeing item 3; ^4^ freeing item 6.

**Table 4 ejihpe-15-00162-t004:** Descriptive statistics, reliability (ω), and association (Pearson *r*) of the SECAB-A(S) scales (*N* = 767).

Variables	*M* (*SD*)	Ω [95% CI]	1.	2.	3.	4.	5.
1. Self-awareness	7.36 (1.34)	0.81 [0.79, 0.83]	_				
2. Self-regulation	6.41 (1.56)	0.84 [0.83, 0.87]	0.60 **	_			
3. Conflict management	7.18 (1.19)	0.73 [0.69, 0.75]	0.41 **	0.41 **	_		
4. Positive relationship	7.04 (1.36)	0.77 [0.76, 0.80]	0.56 **	0.51 **	0.58 **	_	
5. Responsible decision-making	7.17 (1.38)	0.78 [0.76, 0.81]	0.54 **	0.60 **	0.53 **	0.64 **	_

** *p* < 0.001.

**Table 5 ejihpe-15-00162-t005:** Intercorrelation between SECAB-A(S) scales and sociodemographic indicators, satisfaction with affective relations, and personal and academic well-being.

Variables	Age	Gender	Academic Field	KMSS
1. Self-awareness	0.13 *	−0.04	−0.08	0.12 *
2. Self-regulation	0.27 **	−0.03	−0.07	0.19 *
3. Conflict management	0.05	−0.03	−0.07	0.10
4. Positive relationship	0.18 *	0.05	−0.09	0.14 *
5. Responsible decision-making	0.11	−0.03	−0.06	0.20 **

KMSS = Satisfaction with affective relations. * *p* < 0.05, ** *p* < 0.01.

**Table 6 ejihpe-15-00162-t006:** Intercorrelation between SECAB-A(S) scales and personal and academic well-being dimensions.

Variables	Personal Well-Being			Academic Well-Being		
	Emotional	Psychological	Social	Vigor	Dedication	Absorption
1. Self-awareness	0.27 **	0.43 **	0.20 **	0.28 **	0.27 **	0.22 **
2. Self-regulation	0.54 **	0.67 **	0.45 **	0.58 **	0.53 **	0.56 **
3. Conflict management	0.31 **	0.40 **	0.28 **	0.30 **	0.32 **	0.34 **
4. Positive relationship	0.34 **	0.48 **	0.33 **	0.33 **	0.37 **	0.33 **
5. Responsible decision-making	0.33 **	0.46 **	0.32 **	0.30 **	0.40 **	0.38 **

** *p* < 0.01.

**Table 7 ejihpe-15-00162-t007:** Participants perceived SEC (mean and standard deviation) by gender and group differences (independent sample *t*-test).

Variable	Gender		Group Differences			
	Female (*n* = 461)	Male (*n* = 265)				
	*M* (*SD*)	*M* (*SD*)	*Statistic*	*p*	95% CI	*d*
Perceived SEC						
*Self-awareness*	7.40 (1.28)	7.26 (1.46)	1.334	0.182	[−0.07, 0.34]	0.10
*Self-regulation*	6.31 (1.51)	6.59 (1.63)	−2.314	0.021	[−0.51, −0.04]	−0.18
*Conflict management*	7.23 (1.17)	7.07 (1.23)	1.753	0.080	[−0.02, 0.34]	0.14
*Positive relationship*	7.07 (1.29)	6.93 (1.50)	1.332	0.183	[−0.07, 0.35]	0.10
*Responsible decision-making*	7.18 (1.33)	7.12 (1.49)	0.548	0.584	[−0.15, 0.27]	0.04

**Table 8 ejihpe-15-00162-t008:** Participants’ sociodemographic characteristics and perceived SEC (mean and standard deviation) by academic field.

Variable	Academic Field				Group Differences			
	STEM (*n* = 513)		HASS-H (*n* = 219)					
	%	*M* (*SD*)	%	*M* (*SD*)	*Statistic*	*p*	95% CI	*d*
Age		21.83 (5.77)		25.23 (9.49)	5.093 ^a^	<0.001	[2.26, 5.12]	0.48
Gender (Female)	54.0		83.4		0.284 ^b^	<0.001		
Perceived SEC								
*Self-awareness*		7.28 (1.33)		7.57 (1.36)	2.632 ^a^	0.004	[0.07, 0.51]	0.21
*Self-regulation*		6.34 (1.59)		6.58 (1.52)	1.911 ^a^	0.028	[−0.01, 0.51]	0.16
*Conflict management*		7.17 (1.21)		7.19 (1.17)	0.210 ^a^	0.417	[−0.17, 0.22]	−0.01
*Positive relationship*		7.02 (1.37)		7.07 (1.36)	0.395 ^a^	0.346	[−0.18, 0.27]	0.03
*Responsible decision-making*		7.19 (1.39)		7.16 (1.38)	−0.223 ^a^	0.412	[−0.25, 0.20]	−0.03

^a^ Independent samples *t*-test; ^b^ Chi-square test. *Note*. STEM group includes the following education and training fields: agriculture, forestry, fisheries and veterinary sciences, natural sciences, mathematics and statistics, engineering, manufacturing and construction, and ICTs. HASS-H group includes the following education and training fields: education, arts and humanities, social sciences, journalism and information, business sciences, administration and law, health and social protection, and services.

## Data Availability

The datasets generated and analyzed during this study can be found in the *Open Science Framework* repository at Oliveira, S. et al. Dataset of the paper “Mind the (Social and Emotional Competence) Gap to Support Higher Education Students’ Well-Being: Psychometric Properties of the SECAB-A(S)”. OSF, 7 August 2025. Web. https://doi.org/10.17605/OSF.IO/MZYHG.

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
