# Peer review of "Mind the (Social and Emotional Competence) Gap to Support Higher Education Students’ Well-Being: Psychometric Properties of the SECAB-A(S)"

_ejihpe, 2025, doi:10.3390/ejihpe15080162_

Round 1

Reviewer 1 Report

Comments and Suggestions for Authors

Review: Mind the (Social and Emotional Competence) Gap to Support 2 Higher Education Students' Well-Being: Psychometric Proper-3 ties of the SECAB-A(S)

Thank you for the opportunity to review the manuscript entitled “Mind the (Social and Emotional Competence) Gap to Support 2 Higher Education Students' Well-Being: Psychometric Proper-3 ties of the SECAB-A(S)”. I recognize the work involved in writing and preparing a manuscript for submission to a Journal.

The aim of the present study was to evaluate the psychometric properties of the SECAB-A(S), namely its structural, discriminant and concurrent criterion validity, reliability, and multi-group invariance across gender and academic fields. The title is accurate, the abstract describes the main sections of the study, and the keywords are correct. The topic is important, and the conceptual analysis made in the text is quite deep. The most literature consulted is quite current, but the sample is not quite large for what is recommended in an instrument validation study (which is a limitation for your work). I would like to thank the efforts by the authors of the manuscript and congratulate them on the work. Overall, the writing is clear, the goals are well described, the introduction explains the objectives of the study based on the review of the previous literature and the conclusions are properly made and presented. In this sense, the manuscript describes the main study hypotheses considering the previous scientific evidence. The methodology section is well described, and the statistical analyses are well justified. A detailed description of the statistics used, and their relevance is provided. I think the sections on participants and procedure are well explained. The results obtained are presented in an appropriate way, following a narrative and argumentative thread. In addition, an adequate discussion is presented considering previous research, not just the studies in the introduction. Therefore, the manuscript provides significant knowledge of scientific literature and covers existing gaps in the field. I think the following result should be better explained and justified and not overlooked: “This aligns with the view that self-awareness, in isolation, may amplify internal focus without necessarily fostering resolution or change, particularly under stress” (page 7 of the manuscript, lines 740-742). I think you should elaborate on this statement. I think that the limitations section could be completed with some more limitations such as sample size, social desirability bias and the influence of other variables that could be mediating the direction of the results. In addition, it would be interesting to take into account qualitative measures in order to know in detail the responses of the participants. I consider that the practical implications of the study, the
novelty with respect to previous studies and the educational scope of the results obtained should be much better explained. A final section of conclusions should be added. In this regard, a section on conclusions consistent with the evidence and arguments presented should be added. The number of references is adequate and most of the studies are recent and of international impact.

In summary, The study has several strengths (literature review, scope of the research, relevance of the topic) but it also has some weaknesses that the authors should take into account (sample size, type of design, specific cultural and social context of the sample, influence of other variables that may impact on social and emotional competence). I believe that they should describe more clearly what is new and what is the practical relevance of the validated instrument with respect to similar instruments already published.

On a formal level, the manuscript should be better structured, and it would be convenient to reorganize the information by adding new headings, the references comply with the rules of the Journal and the DOI is added. Tables and figures are relevant and well presented. They also present information relevant to the study. The work is ambitious, and the results confirm most of the hypotheses and the relevance and potential of the work is therefore recognized, but this Reviewer considers that several changes are needed to the manuscript is publishable. Some are of content, and other formal. In this regard, I consider that the manuscript I consider that the manuscript can continue with the review process. Finally, I wish the authors the best in continuing this line of research.

Best wishes to the Authors.

Author Response

Reviewer #1

Thank you for the opportunity to review the manuscript entitled “Mind the (Social and Emotional Competence) Gap to Support Higher Education Students' Well-Being: Psychometric Properties of the SECAB-A(S)”. I recognize the work involved in writing and preparing a manuscript for submission to a Journal.

The aim of the present study was to evaluate the psychometric properties of the SECAB-A(S), namely its structural, discriminant and concurrent criterion validity, reliability, and multi-group invariance across gender and academic fields.

Thank you very much for your encouraging feedback and for taking the time to review our paper in such detail. We found your suggestions very pertinent and constructive for strengthening our paper. We tried to elaborate our paper by introducing all the information and amendments you suggested. We hope we were able to respond to your concerns and offset our paper’s limitations. We know how time consuming it is to make revisions, and we really appreciate your availability.

Specific Suggestions:

  1. The title is accurate, the abstract describes the main sections of the study, and the keywords are correct. The topic is important, and the conceptual analysis made in the text is quite deep.

Answer 1: Thank you very much for your positive feedback.

  1. The most literature consulted is quite current, but the sample is not quite large for what is recommended in an instrument validation study (which is a limitation for your work).

Answer 2: Thank you for your valuable feedback. Regarding the sample size, we would like to clarify that our study followed established guidelines for instrument validation. Specifically, we ensured a participant-to-parameter ratio of 10:1, as recommended by Kline (2016). Additionally, a-priori power analysis for the CFA model (power = .80, p = .05, RMSEA < .05), following Moshagen & Bader (2024), indicated that a minimum sample size of 182 participants would be sufficient to reliably test the structural model. Our final sample consisted of 767 participants, which substantially exceeds both commonly recommended thresholds (300-500 participants) and the required sample size highlighted by the power analysis.

Nonetheless, we acknowledge that the use of online data collection and convenience sampling introduces potential limitations regarding representativeness and generalizability. We have addressed and clarified these points in the Limitations section of our manuscript.

  1. I would like to thank the efforts by the authors of the manuscript and congratulate them on the work. Overall, the writing is clear, the goals are well described, the introduction explains the objectives of the study based on the review of the previous literature and the conclusions are properly made and presented. In this sense, the manuscript describes the main study hypotheses considering the previous scientific evidence. The methodology section is well described, and the statistical analyses are well justified. A detailed description of the statistics used, and their relevance is provided. I think the sections on participants and procedure are well explained. The results obtained are presented in an appropriate way, following a narrative and argumentative thread. In addition, an adequate discussion is presented considering previous research, not just the studies in the introduction. Therefore, the manuscript provides significant knowledge of scientific literature and covers existing gaps in the field.

Answer 3: Thank you very much for your thoughtful comments and encouraging feedback.

  1. I think the following result should be better explained and justified and not overlooked: “This aligns with the view that self-awareness, in isolation, may amplify internal focus without necessarily fostering resolution or change, particularly under stress” (page 7 of the manuscript, lines 740-742). I think you should elaborate on this statement.

Answer 4: Thank you very much for your suggestion. We have, accordingly, elaborate on this finding and, particularly, on the statement you highlighted. Please see these changes on p. 22-23.

  1. I think that the limitations section could be completed with some more limitations such as sample size, social desirability bias and the influence of other variables that could be mediating the direction of the results. In addition, it would be interesting to take into account qualitative measures in order to know in detail the responses of the participants.

Answer 5: Thank you for your concern. We have further developed the Limitations section, adding more information regarding the sampling methodology, biases and possible impact of unmeasured variables. Please find these changes on p.22-23.

  1. I consider that the practical implications of the study, the novelty with respect to previous studies and the educational scope of the results obtained should be much better explained.

Answer 6: Thank you for your feedback. Following your suggestion, we have further developed the Study impact section, emphasizing the study contributions, novelty and practical implications.

  1. A final section of conclusions should be added. In this regard, a section on conclusions consistent with the evidence and arguments presented should be added.

Answer 7: Thank you for your suggestion. We agree and have added a conclusion section at the end of the manuscript to sum up the main take-out ideas and contribution of the paper.

  1. The number of references is adequate and most of the studies are recent and of international impact.

Answer 8: Thank you for the feedback.

9. In summary, The study has several strengths (literature review, scope of the research, relevance of the topic) but it also has some weaknesses that the authors should take into account (sample size, type of design, specific cultural and social context of the sample, influence of other variables that may impact on social and emotional competence).

Answer 9: Thank you for your suggestion. We took it into account, and we have further developed the Limitations section. Please find these changes on pp. 22-23.

  1. I believe that they should describe more clearly what is new and what is the practical relevance of the validated instrument with respect to similar instruments already published.

Answer 10: Thank you for your concern. We have re-written the Study impact section to more clearly describe the novelty and practical relevance of the SECAB-A(S).

  1. On a formal level, the manuscript should be better structured, and it would be convenient to reorganize the information by adding new headings, the references comply with the rules of the Journal and the DOI is added. Tables and figures are relevant and well presented. They also present information relevant to the study. The work is ambitious, and the results confirm most of the hypotheses and the relevance and potential of the work is therefore recognized, but this Reviewer considers that several changes are needed to the manuscript is publishable. Some are of content, and other formal. In this regard, I consider that the manuscript can continue with the review process. Finally, I wish the authors the best in continuing this line of research.

Answer 11: Thank you very much for your feedback. We have, accordingly added new sub-headings following the journal’s template to better organize the information, namely in the Introduction and Discussion sections.

Best wishes to the Authors.

Thank you very much.

Reviewer 2 Report

Comments and Suggestions for Authors

The aim of this study was to evaluate the psychometric properties of the SECAB-A(S) including its structural, discriminant and concurrent criterion validity, reliability, and multi-group invariance across gender and academic field. And, specifically to confirm whether this instrument would be valid and reliable within higher education.

A strength of this manuscript is the in-depth analysis of the data, thorough literature review and substantiation of the problem and reason for the study, and strong conclusions.

A weakness or limitation of the study was the population was limited due to a convenience sample and primarily included Portuguese students.  

Thank you for this well-written manuscript. I believe it offers much to the existing knowledge base out there on SEC and I value the applicability to higher education.

Author Response

Reviewer #2

The aim of this study was to evaluate the psychometric properties of the SECAB-A(S) including its structural, discriminant and concurrent criterion validity, reliability, and multi-group invariance across gender and academic field. And, specifically to confirm whether this instrument would be valid and reliable within higher education.

A strength of this manuscript is the in-depth analysis of the data, thorough literature review and substantiation of the problem and reason for the study, and strong conclusions.

A weakness or limitation of the study was the population was limited due to a convenience sample and primarily included Portuguese students.

Thank you for this well-written manuscript. I believe it offers much to the existing knowledge base out there on SEC and I value the applicability to higher education.

We would like to sincerely thank you for the time and effort dedicated to reviewing our manuscript, as well as for your constructive and positive feedback. We greatly appreciate your recognition of the thorough data analysis, comprehensive literature review, and the relevance and clarity of our conclusions, as well as your acknowledgment of the study’s applicability within higher education.

We also thank you for highlighting the limitation regarding the sampling procedure. We agree with your assessment. In response to your suggestion, we have revised and strengthened the limitations section of the manuscript to provide a more detailed discussion of the constraints associated with the use of a convenience sample. We also clarified that, while our sample predominantly consisted of Portuguese students, this was intended as the instrument was specifically adapted to the Portuguese cultural and educational context. Nevertheless, we recognize the value of further test the instrument in other higher education students’ population with the appropriate cultural and linguistic adaptations – namely in other countries with Portuguese as official language (e.g., Brazil). Once again, we thank you for your valuable input, which has contributed to the improvement of our work.

Reviewer 3 Report

Comments and Suggestions for Authors

This is an interesting paper on an important topic. The key issues are 1. the level of scholarship in the article. Please see the attached marked-up copy that indicates where further authority for statements is required and 2. the clarity of expression.

Please see the attached marked up copy for additional feedback.

Author Response

Reviewer #3

This is an interesting paper on an important topic. The key issues are 1. the level of scholarship in the article. Please see the attached marked-up copy that indicates where further authority for statements is required and 2. the clarity of expression. Please see the attached marked up copy for additional feedback.

We would like to sincerely thank the reviewer for the thoughtful and constructive feedback. We have carefully addressed all the suggestions and comments provided.

To strengthen the scholarly foundation of the manuscript, we have incorporated additional citations throughout the text to support key claims and arguments. We have also included further references to ensure comprehensive coverage and a more robust mapping of the evidence in the field.

In response to concerns regarding clarity of expression, we have undertaken a thorough revision of the manuscript to enhance readability and precision. Furthermore, following your recommendation, we have added a dedicated conclusion section to provide a concise synthesis of the key points and implications of our work.

All revisions have been clearly marked in the manuscript for ease of reference.

Once again, we are deeply grateful for your insightful comments, generous guidance, and encouragement. We appreciate the time and effort you invested in reviewing our manuscript and helping us improve its quality.

Round 2

Reviewer 3 Report

Comments and Suggestions for Authors

Thank you for the opportunity to review this article. I enjoyed reading it and thought it was a well-conceptualised, written and researched piece. Student wellbeing is a critical component of student success and it is therefore very important to be able to measure and understand its dynamics. I defer to methods experts in terms of reviewing the Part 2 Materials and Methods section. However, as an academic in this field I believe the article makes an important contribution to the available literature - especially in terms of the novel measure developed which is well-justified. The depth and breadth of the scholarship considered in the analysis shows a high level of engagement with the literature and synthesis of available sources. The discussion of the results was helpful and informative. It was good to see a level of caution in the interpretation process. The study's limitations were openly acknowledged. 

In relation to improvements that could be made:

  • more thorough referencing of the relevant SDT literature is warranted.
  • perhaps the contentious nature, and various definitions, of 'emotional intelligence' could be better acknowledged.
  • when talking about well-being research in different disciplines eg lines 217 - 272 the significant body of scholarship in the discipline of law seems to have been overlooked. This should be included for a complete picture.

Author Response

Thank you for the opportunity to review this article. I enjoyed reading it and thought it was a well-conceptualised, written and researched piece. Student wellbeing is a critical component of student success and it is therefore very important to be able to measure and understand its dynamics. I defer to methods experts in terms of reviewing the Part 2 Materials and Methods section. However, as an academic in this field I believe the article makes an important contribution to the available literature - especially in terms of the novel measure developed which is well-justified. The depth and breadth of the scholarship considered in the analysis shows a high level of engagement with the literature and synthesis of available sources. The discussion of the results was helpful and informative. It was good to see a level of caution in the interpretation process. The study's limitations were openly acknowledged.

Thank you very much for your positive feedback and for taking the time to review our paper. We understand how time-consuming it can be to make revisions, and we appreciate your availability.

In relation to improvements that could be made:

  • more thorough referencing of the relevant SDT literature is warranted.

Thank you very much for your concern. We have further developed the paragraph to better explain and support the link between SDT and the SEL framework. We have also added new references to sustain our arguments (e.g., Beachboard et al., 2011; Hagenauer et al., 2018, Kurdi et al., 2021).

  • perhaps the contentious nature, and various definitions, of 'emotional intelligence' could be better acknowledged.

Thank you very much for your suggestion. Even though an in-depth analysis of the contentious nature and various definitions of emotional intelligence falls outside the focus of our paper, we agree with the reviewer that, due to the ongoing debate, it would be important to operationalize the construct and present the definition used in our paper – Emotional Intelligence Theory (Salovey & Mayer, 1990). We have added that information early to the paper.

  • when talking about well-being research in different disciplines eg lines 217 - 272 the significant body of scholarship in the discipline of law seems to have been overlooked. This should be included for a complete picture.

Thank you very much for pointing this out. As it was not our goal to study a particular group of students, we chose to characterize the two major academic fields (STEM and HASS-H fields) instead of covering literature on specific disciplines (e.g., medical, law, nursing students). Nevertheless, we understand that this clustering may not have been made clear in the introduction, so we have added this information in sections 1.2.2.1 and 1.2.2.2, and added a reference specifically referring to law students’ strains (James, 2011).